# Metabolic Effects of CCN5/WISP2 Gene Deficiency and Transgenic Overexpression in Mice

**DOI:** 10.3390/ijms222413418

**Published:** 2021-12-14

**Authors:** Tara Alami, Jun-Li Liu

**Affiliations:** 1MeDiC Program, The Research Institute of McGill University Health Centre, Montreal, QC H4A 3J1, Canada; tara.alami@mail.mcgill.ca; 2Division of Endocrinology and Metabolism, Department of Medicine, McGill University, 1001 Blvd Decarie, Montreal, QC H4A 3J1, Canada

**Keywords:** matricellular proteins, knockout mice, transgenic overexpression, diet-induced obesity, pancreatic islets, fibrosis, insulin sensitivity, aP2 promoter

## Abstract

CCN5/WISP2 is a matricellular protein, the expression of which is under the regulation of Wnt signaling and IGF-1. Our initial characterization supports the notion that CCN5 might promote the proliferation and survival of pancreatic β-cells and thus improve the metabolic profile of the animals. More recently, the roles of endogenous expression of CCN5 and its ectopic, transgenic overexpression on metabolic regulation have been revealed through two reports. Here, we attempt to compare the experimental findings from those studies, side-by-side, in order to further establish its roles in metabolic regulation. Prominent among the discoveries was that a systemic deficiency of CCN5 gene expression caused adipocyte hypertrophy, increased adipogenesis, and lipid accumulation, resulting in insulin resistance and glucose intolerance, which were further exacerbated upon high-fat diet feeding. On the other hand, the adipocyte-specific and systemic overexpression of CCN5 caused an increase in lean body mass, improved insulin sensitivity, hyperplasia of cardiomyocytes, and increased heart mass, but decreased fasting glucose levels. CCN5 is clearly a regulator of adipocyte proliferation and maturation, affecting lean/fat mass ratio and insulin sensitivity. Not all results from these models are consistent; moreover, several important aspects of CCN5 physiology are yet to be explored.

## 1. Introduction

In recent years, we have reported a significant induction by IGF-1 of the expression of matricellular protein CCN5/WISP2 and its in vitro effects on promoting pancreatic β-cell proliferation and survival [1,2]. In 2017, we reviewed how CCN5 might be involved in the molecular regulation of pancreatic islets and therefore in metabolic activities [3]. Since then, there has been a significant development in gene deletion and transgenic overexpression, which has warranted a revisit [4,5].

## 2. General Overview of CCN/WISP Family of Matricellular Proteins

The CCN/WISP family, which consists of six matricellular proteins, regulates development, cell adhesion and proliferation, extracellular matrix (ECM) remodeling, inflammation, and tumorigenesis. Most of these proteins contain four functional domains (Figure 1): the IGF-BP domain, which has a sequence homology similar to the six classic IGF-BPs, but confers only <1% of affinity to insulin-like growth factors (IGFs) [6]; the von Willebrand factor type C repeat (VWC) associated with the ECM; the thrombospondin type I repeat (TSP-1) involved in attachment to integrins; and the cysteine-rich C-terminal repeat (CT, or heparin-binding domain) associated with ligand dimerization and receptor binding [7,8,9,10,11]. Overall, they share ~50% of amino acid sequences, including 38 cysteines and an N-terminal signal peptide destined for exocytotic secretion, and are believed to form multimeric complexes. Acting either as matricellular proteins [12] or, as we and others have proposed, *growth factors* through their interactions with cognate receptors [12,13,14], CCNs regulate the expression and activities of growth factors, cytokines, and matrix metalloproteinases (MMPs) [15].

While searching for novel growth factors to promote the proliferation and survival of pancreatic islets, we revealed the expression of CCN5/WISP2 (cellular communication network factor 5, Wnt inducible signaling protein 2) in resting pancreatic β-cells and its robust inductions by IGF-1, and further reported that recombinant human protein (rhCCN5) promotes mouse β-cell proliferation and survival in vitro [1,2]. In a previous article, we have reviewed the role of CCN5 in the regulation of pancreatic islet function and metabolic activities in general [3].

Before assessing its normal physiology, it is necessary to know where (i.e., in which tissues and cells) CCN5 is normally expressed and how the expression might be regulated (e.g., by obesity and diabetes). It has been reported that CCN5 is expressed throughout murine embryonic development in most organs and tissues [17,18]. According to the Human Protein Atlas (www.proteinatlas.org/ENSG00000064205-WISP2/tissue, accessed on 9 December 2021), CCN5 is a secreted plasma protein that is highly expressed in the proximal gastrointestinal tract, male and female reproductive tissues (testis, uterine), and adipose tissue. Perhaps more accurately, Genotype-Tissue Expression (GTEx) project is an ongoing effort to build a comprehensive resource to study tissue-specific gene expression and regulation. From there, an expression profile of human CCN5/WISP2 can be retrieved, as shown in Figure 2, which highlights CCN5 expression in blood vessels and subcutaneous adipose tissues. However, these results have not been studied in detail and peer reviewed; the results are not all consistent to each other.

Immunohistochemical (IHC) analysis of mouse **heart** tissue revealed CCN5 expression throughout the cytoplasm of the ventricular myocardium, in the atria and in the valves, and in most nuclei in the myocardium. In the **lung**, CCN5 was observed in the cytoplasm of alveolar and bronchiolar epithelial cells, endothelial cells, and smooth muscle cells, and also in the nuclei of many cell types [17]. In both cases, however, the image resolution was insufficient to clearly define a *nuclear* localization. All layers of mouse **stomach** wall and duodenum also displayed CCN5 staining in the cytoplasm. In mice, using IHC, CCN5 is expressed in the cytoplasm of acinar cells through the **pancreas**; but there was no nuclear staining [17]. In humans, analysis of **adipose** protein secretome highlights CCN5 as a key regulator of obesity and ECM interactions [19]. Indeed, CCN5 is one of the only five genes up regulated in obese women. As a novel adipokine, CCN5 expression was significantly increased in human obesity and insulin resistance [20].

Furthermore, CCN5 is highly expressed in the **testis,** with very high expression in peritubular cells and Leydig (testosterone-producing) cells. Numerous genes are expressed in mouse testis, yet most of them have not been studied for their involvement in spermatogenesis and sperm function. Analyzing the phenotype of CCN5-knockout mice might provide insight into the function of CCN5 in male reproduction. In rat and mouse **ovaries**, CCN5 is expressed in all cell types including stromal cells, thecal cells, granulosa cells, and oocytes [17]. Previous studies have revealed that CCN5 is normally expressed in the rat **uterus** in the smooth muscle, glandular epithelium, and the endometrium [21], and rat arterial smooth muscle and endothelium [17]. In the human fetus at 5 months, low levels of CCN5 staining were detected in testicular Leydig cells, the uterus, ovarian stroma, fallopian tube, and epididymis [18]. In addition to these early observations, a detailed examination of CCN5 expression in various systems can also be found in Grunberg et al. and Twigg et al.’s reviews on the topic [22,23].

## 3. The Effects of a Systemic CCN5/WISP2 Gene Deficiency

As expected from previous reports, CCN5/WISP2 gene deficiency has been associated with mild obesity, insulin resistance, hyperglycemia, and lipotoxic cardiomyopathy [4]. In a separate report using this model, CCN5 expression is not required for normal bone formation [24]. Although these findings support a role for endogenous CCN5 in the regulation of metabolic homeostasis, the CCN5-mediated mechanism of action, including in pancreatic islets, has yet to be elucidated.

### 3.1. Adipocyte Hypertrophy, Increased Adipogenesis, and Mild Obesity

First, CCN5-knockout mice fed the normal chow diet (NCD) have been shown to exhibit mild obesity in comparison to wild-type littermates, despite no significant change in food intake [4]. When fed a high-fat diet (HFD), they did exhibit increased food intake together with obesity vs. wild-type littermates [4]. These changes are consistent with the notion that CCN5 is thought to partially prevent obesity by inhibiting the TGF-β signaling pathway, of which Smad3 is a downstream mediator. Indeed, Smad3-deficient mice are resistant to HFD-induced obesity and diabetes [25]. The latter was previously confirmed in Smad3-deficient mice fed a HFD, which exhibited increased whole-body glucose uptake, improved insulin sensitivity, and decreased fasting insulin and glucose levels [26]. HFD-fed, Smad3-deficient mice also gained less weight than their wild-type littermates and were protected from ectopic lipid accumulation in the liver [26]. Activation of the TGF-β/Smad3 pathway is reported to promote a leaner phenotype via Smad3′s regulation of PPARγ coactivator-1α (PGC-1α), which regulates metabolic genes [26]. Indeed, TGF-β represses PGC-1α in a Smad3-dependent manner, and the deletion of Smad3 gene relieves this inhibition [26]. Thus, the lack of CCN5 signaling may result in mild obesity, possibly through **elevated TGF-β/Smad3** signaling activities. Future experiments will be required to directly establish this mechanism.

Alongside obesity, the mass of subcutaneous and perirenal white adipose tissues (sWAT and pWAT) and of the heart was significantly increased in NCD-fed CCN5-knockout vs. wild-type mice [4]. The increase in the fat mass may be accounted for by the notable increases in the expression of **adipogenic genes** and transcription factors, such as sterol regulatory element-binding protein (SREBP1), CCAAT/enhancer-binding protein alpha (C/EBPα), peroxisome proliferator-activated receptor-gamma (PPARγ), and activating protein 2 (aP2) [4]. When activated, transcription factors C/EBPα and PPARγ activate the expression of genes that induce an adipocytic phenotype [27]. In the meantime, CCN5 knockout may have removed the inhibition of adipogenic differentiation in mesenchymal precursor cells (MPCs), since CCN5 has been shown to maintain MPCs in an undifferentiated state through activation of the canonical WNT signaling pathway [4]. With the rise in adipogenesis, CCN5-knockout mice also exhibited adipocyte hypertrophy in comparison to their wild-type littermates [4]. These results further suggest the physiological suppression of adipogenesis by normal CCN5 expression. Together, CCN5 knockout would result in increased adipocyte differentiation, activated TGF-β/Smad3 pathway, and the expression of adipogenic genes, which all contribute to mild obesity.

Cellular hypertrophy was also exhibited in the heart of NCD-fed CCN5-knockout mice, with an increase in the mRNA expression of the hypertrophy-associated genes myosin heavy chain beta (β-MHC) and skeletal actin [4]. HFD-fed knockout mice exhibited further hypertrophy [4]. We speculate that a similar molecular mechanism may have caused cellular hypertrophy in both adipocytes and cardiomyocytes. In fact, CCN5 staining was detected in cardiomyocytes [17]; its expression in fibroblasts of the left ventricle was induced by myocardial infraction in mice [28]. In future studies, a cardiomyocyte- vs. adipocyte-specific knockout of CCN5 could help to determine whether the same hypertrophy mechanism applies to both adipocytes and cardiomyocytes.

### 3.2. Increased Lipid Accumulation and Fibrosis in the Heart

While NCD-fed CCN5-knockout mice exhibited lipid accumulation in the heart, they also showed an increase in the mRNA levels of lipid *oxidation*-associated genes (CPT1β and MCAD) and glycerol-3-phosphate acyltransferase (GPAT), which is associated with triglyceride synthesis [4]. Although the increase in lipid oxidation-related genes does not align with lipid accumulation, the latter may be explained by the increase in cell size due to hypertrophy and impaired glucose metabolism, likely leading to increased lipid oxidation and utilization. As such, triglyceride synthesis most likely overrides lipid oxidation, resulting in a net *accumulation* of fat. Perhaps associated with that, NCD-fed CCN5-knockout mice also exhibited fibrosis in the heart, particularly in interstitial and perivascular areas. This was confirmed through collagen deposition in those areas and an increase in the expression of fibrosis-related genes (TGF-β1, Col1, and α-SMA) and inflammation-related genes (F4/80/CD3 and CD11c) [4]. Interestingly, CCN5 has been shown to inhibit TGF-β activation in myofibroblasts and the level of CCN5 was reduced in individuals with fibrotic heart failure [29]. The lack of TGF-β inhibition in the face of CCN5 deficiency may thus trigger an immune response through the expression of proinflammatory genes and facilitate fibrosis in the heart. Indeed, obesity is linked to the activation of pro-fibrotic signaling through the renin-angiotensin-aldosterone pathway, TGF-β, and oxidative stress [30]. Further, cardiac fibrosis is strongly associated with metabolic dysfunction, including obesity [30,31].

Hence, CCN5 gene deficiency led to lipid accumulation in the heart, despite an apparent increase in lipid oxidation too. Cardiomyocyte hypertrophy may have occurred through a similar mechanism as in adipose tissues, involving increased differentiation of MPCs. Moreover, CCN5 knockout caused cardiac fibrosis and collagen deposition. The lack of CCN5 gene expression likely allowed the induction of a pro-fibrotic pathway due to elevated activities of the TGF-β/Smad3 signaling. On the other hand, whether this pathway is also involved in the induction of fibrosis in *adipose tissues* remains to be determined.

To further study the cellular mechanism, cardiomyocytes from NCD-fed CCN5 knockout-mice were isolated which exhibited mild insulin resistance. The latter was confirmed by reduced phosphorylation of AKT1 and GSK-3β following insulin administration relative to cardiomyocytes isolated from wild-type mice [4]. This was consistent with the mild hyperglycemia and hyperinsulinemia phenotypes and suggests that cardiomyocytes may have switched to more lipid catabolism for energy, thus increasing the expression of lipid oxidation-associated genes. The combination of lipid accumulation, increased lipid oxidation, mild insulin resistance, and cardiac fibrosis may have all contributed to systolic and diastolic dysfunctions of the heart [4]. Systolic dysfunction is associated with an abnormal left ventricular ejection fraction, which has been reported to be increased in obese individuals [31]. Diastolic dysfunction is proportionally associated with obesity, too, and is characterized by prolonged left ventricular relaxation, decreased blood flow velocity through the mitral valve and E/A ratio, and elevated ventricle filling pressure [31]. 

### 3.3. Mild Hyperglycemia, Hyperinsulinemia and Cardiac Dysfunction

Perhaps due to the mild obesity, lipid accumulation, and increased lipid oxidation, CCN5-knockout mice fed an NCD also showed mild hyperinsulinemia, hyperglycemia, and insulin resistance [4]. On the other hand, HFD-fed CCN5 knockout mice exhibited increased water intake, consistent with a mild diabetic phenotype, together with higher blood glucose and insulin levels [4]. Interestingly, while fasting glucose levels were notably higher in NCD-fed CCN5 knockout mice, *HFD-fed* CCN5-knockout mice did not exhibit such an increase vs. wild-type mice [4].

As summarized in Table 1, under normal chow diet, CCN5 knockout caused mild obesity, adipocyte hypertrophy, mild hyperglycemia, mild hyperinsulinemia, cardiac hypertrophy, lipid accumulation, fibrosis, and cardiac deficiency, and decreased water intake. Further effects of a **prolonged HFD** (24 weeks) include mild obesity, further adipocyte hypertrophy, frank diabetes, insulin resistance, glucose intolerance, hyperinsulinemia, increased food and water intake, and further defect in cardiac function, perhaps due to lipid accumulation and cellular hypertrophy.

**In summary**, a systemic deficiency of CCN5 gene expression caused adipocyte hypertrophy, increased adipogenesis, and lipid accumulation, resulted in insulin resistance and glucose intolerance, which are further exacerbated upon HFD feeding. In addition, CCN5 deficiency caused cardiac fibrosis, cardiomyocyte hypertrophy, and lipid accumulation, leading to significant deficits in cardiac function. With little increase or no change in food intake, we would expect decreased energy expenditure in these mice, which has not been measured. Based on our reports, we also expect changes in pancreatic β-cell mass or function. It should be noted that these reports have never specified *the sex* of the animals [4], while our preliminary results clearly differ and show significant sexual dimorphic patterns in these mice (*J.L. Liu* et al. Unpublished observations). Nevertheless, current observations indicate that normal endogenous expression of CCN5 gene suppresses adipogenesis, but promotes/maintains cellular division, insulin sensitivity, and cardiac function.

## 4. Differential Roles Played by CCN2, CCN3, and CCN4 in Obesity, Fibrosis, and Pancreatic Islets

Another member of the family, **CCN3/NOV**, has also been reported to be an adipocytokine involved in obesity-associated insulin resistance (9). Excess CCN3 is associated with the induction of obesity, insulin resistance, and impaired adipocyte differentiation [23]. In contrast to CCN5, the complete opposite changes were caused by CCN3 gene deletion, i.e., essentially normal with a chow diet, but *decreased* body weight and fat mass with an HFD, *decreased* adipocyte size, and significantly *increased* energy expenditure, glucose tolerance, and insulin sensitivity/signaling [32]. Increased energy expenditure seems to be the primary cause of those changes. The role of endogenous CCN3 thus would contrast with what was concluded above on CCN5, i.e., to increase adipocyte proliferation but limit adipocyte expansion (hypertrophy), adiposity, and insulin responsiveness. While surprising, these opposite results obtained through similar knockout on different CCN genes support the notion that CCN5 (due to its lack of CTD, Figure 1) may serve as a functional antagonist (or dominant negative) to other CCN isoforms.

In pancreatic islets, CCN3 expression is upregulated in animal models of insulin resistance. CCN3 overexpression in vitro has been reported to disrupt β-cell function and proliferation, reducing cAMP levels, and decreasing glucose oxidation, thereby impairing insulin secretion [33]. However, the aforementioned CCN3 knockout caused no change in pancreatic insulin content, serum insulin level and insulin response [32]. Moreover, excess CCN3 induces obesity, insulin resistance, and disrupt pancreatic islets; this is similar to CCN5 knockout, demonstrating the contrasting roles played by CCN3 and CCN5 again.

**CCN4/WISP1 is a circulating factor** that stimulates the proliferation of mouse and human β-cells [34]. It is well known that β-cell proliferation is more active in young animals; higher levels of trophic factors are suspected in the blood. Using an antibody array, Fernandez-Ruiz et al. screened sera taken from 2- and 20-week-old mice focusing on extrinsic factors to the islets. They discovered CCN4 was six-fold higher in young vs. adult mice. Similarly, human children have three-fold higher level of serum CCN4 than adults. They then compared possible change in the rate of β-cell proliferation in vivo in knockout mice. As a result, CCN4 gene-deficient young mice exhibited a 43% decrease in β-cell proliferation (Ki67-labeling) vs. wild-type littermates. On the other hand, recombinant CCN4 injections (1 mg/kg, 3 days) increased β-cell proliferation by 1.8-fold in knockout mice vs. saline-injected animals. The adenovirus-mediated expression of CCN4 further increased the proliferation of endogenous β-cells in diabetic mice. Finally, recombinant CCN4 protein promoted primary islet proliferation ex vivo at a dose of 0.5 µg/mL. The induction of β-cell replication was dependent on Akt activation [34].

Another family member, **CCN2/CTGF**, is essential for embryonic development of pancreatic islets [9,35,36,37,38]. From early embryonic age (E14.5), the CCN2 gene is highly expressed in pancreatic ducts, vascular endothelium, and mesenchyme, as well as at lower levels in immature β-cells. However, its expression is effectively switched off in β-cells from E18.5 up to postnatal P1, before the onset of postnatal expansion and maturation of β-cells [38]. Hence, CCN2 may not be a growth factor for *adult* β-cells. CCN2-knockout mice die after birth and exhibit altered islet morphology with diminished β/α cell ratio [37,38]; the heterozygous knockout mice do survive, but exhibit a *transient* β-cell deficit at an early age [35]. Combinatorial cell-specific knockout further demonstrated concerted actions of vessel- and β-cell derived CCN2 in the embryo, i.e., CCN2 knockout in endothelial cells decreased islet vascularity; and β-cell specific knockout decreased β-cell proliferation. In contrast, the tetracycline-induced β-cell-specific *overexpression* of CCN2 stimulated the replication of immature β-cells [37]. Thus, CCN2 promotes new islet formation from pancreatic ducts, immature β-cell expansion, islet vascular development, and the differentiation of endocrine progenitors to mature β-cells [35,37]. Based on our studies, CCN5 plays a similar, pro-islet role.

In contrast to CCN5, however, **CCN2 is pro-fibrotic**. Fibrosis is an excess deposition of extracellular matrix components including collagens causing the overgrowth, hardening, and/or scarring of various tissues. Normal tissues can be replaced with permanent scars. It is usually a result of chronic inflammatory reactions to persistent infections, autoimmune reactions, allergic responses, chemical insults, radiation, or other injuries [39]. Fibrosis following a heart attack, and the formation of scar tissues can impair the pumping function. In this aspect, there seems to be a clear isoform specificity, in that CCN5, together with CCN1 and CCN3, are anti-fibrotic, while CCN2 and CCN4 promote fibrosis [40]. Indeed, CCN5 has been known to *prevent* fibrosis and can even reverse established cardiac fibrosis [29]. In the failing human heart, CCN5 protein level is decreased by 24%. Using adenovirus-mediated gene transfer, CCN5 overexpression reversed cardiac fibrosis established in mouse heart failure by decreasing the generation of myofibroblasts in the myocardium. Further experiments demonstrated that CCN5 acts by inhibiting TGF-β signaling, which normally promotes the endothelial–mesenchymal transition and consequent transdifferentiation into myofibroblasts; and by directly causing apoptosis in some myofibroblasts.

CCN2 seems to be pro-fibrotic. Although not required for TGF-β-induced collagen or α-SMA expression in fibroblasts, CCN2 expression is necessary for the recruitment of pericyte-like cells in dermal fibrogenesis [41]. Indeed, CCN2 is specifically upregulated in fibrosis and wound healing. In the heart, cell-specific knockout confirmed that cardiomyocytes (vs. fibroblasts) are the predominant source of CCN2 production (secretion). After inducing cardiac fibrosis by administering angiotensin II for one week, CCN2 expression was strongly induced in cardiomyocytes of wild-type but not CCN2-knockout mice [42]. On the other hand, fibroblasts with a CCN2 deletion lost the pro-fibrotic property and failed to induce fibrosis. Hence, fibroblast derived CCN2 is necessary for fibrosis induction and collagen production. Although both cardiomyocytes and fibroblasts secrete CCN2, only those derived from fibroblasts can strongly induce α-SMA expression and thereby fibrosis. Previously, the differential expression and opposing effects of CCN2 and CCN5 on the development of cardiac hypertrophy and fibrosis has been reported [43]. Thus, CCN5 seems to be synergistic with CCN2 in terms of pancreatic islets, but plays an opposite role in heart fibrosis (summarized in Table 2).

## 5. Metabolic Effects of Adipocyte-Specific and Systemic Overexpression of CCN5

In clear contrast to the aforementioned findings from CCN5-knockout mice, adipocyte-specific CCN5 overexpression caused increases in lean body mass and insulin sensitivity, and cardiac hyperplasia [5]. Major findings have been summarized in Table 1 and reviewed previously; we try not to repeat them in detail [22].

The human adipocyte fatty acid-binding protein (aP2) promoter was chosen to generate aP2-CCN5 transgenic mice because it can be activated by PPARγ and it has been used to specifically overexpress the canonical WNT ligand WNT10b in adipose tissues [5,27]. Further, aP2 is predominantly expressed in adipose tissues and is regulated during adipocyte differentiation. aP2-deficient mice exhibited abnormal glucose and lipid metabolism, suggesting a key role for endogenous aP2 expression in those processes [44]. In addition, the aP2 promoter is also capable of inducing gene expression in macrophages and cardiomyocytes [44]. In this report, the aP2-CCN5 mice exhibited a five-fold increase in the serum level of CCN5, supporting its endocrine secretion [5]. As such, CCN5 overexpression is not only restricted to adipose tissues, but also in systemic circulation, allowing for the assessment of metabolic and physiological effects in other tissues too.

Transgenic aP2-CCN5 mice overexpressing CCN5 exhibited no overall change in body weight when fed either a low-fat or high-fat diet (LFD or HFD) vs. wild-type littermates [5]. However, they exhibited an increased percentage of lean body mass and reduced percentage of fat mass. The change in lean/fat mass was associated with an increase in the mass of brow fat, heart, skeletal muscles, and epididymal and retroperitoneal, white adipose tissues (eWAT and rWAT) in both LFD- and HFD-fed transgenic mice [5]. In HFD, the increases in lean body mass, oxygen consumption, and whole-body energy expenditure were closely associated with an increase in food intake [5].

When challenged with a period of HFD, aP2-CCN5 transgenic mice exhibited improved insulin sensitivity and decreased fasting glucose levels compared to wild-type mice [5]. The lower glucose levels were associated with decreased hepatic glucose production, as confirmed by a pyruvate tolerance test. In fact, the insulin sensitivity in HFD-fed transgenic mice was improved to a similar level as LFD-fed wild-type mice [5]. GLUT4 protein levels were increased in skeletal muscles, subcutaneous WAT, and eWAT of transgenic mice relative to wild-type, providing a mechanism for improving glucose tolerance and insulin sensitivity. Further, transplantation of subcutaneous adipose tissues from the aP2-CCN5 transgenic mice into wild-type recipients caused improved glucose tolerance vs. sham-operated animals [5]. It indicates that the transgenic affected adipose tissues improved insulin sensitivity in the recipient mice.

To confirm an increased level of lipogenesis, the serum levels of metabolically beneficial fatty acid esters of hydroxy fatty acids (FAHFAs) and their isomers, palmitic acid–hydroxy stearic acids (PAHSAs), were measured. Serum levels of 13/12- and 5-PAHSA were increased in HFD-fed transgenic vs. wild-type mice [5]. The increase in lipogenesis markers, GLUT4 and ChREBP, and the levels of FAHFAs/PAHSAs similarly support an improved insulin sensitivity in transgenic mice fed an HFD [5]. In wild-type mice transplanted with transgenic adipose tissues, total PAHSA levels were also significantly increased [5]. PAHSAs have been reported to promote the secretion of glucagon-like peptide 1 (GLP-1) from enteroendocrine cells and the maturation of pancreatic β-cells [45]. The increased PAHSA level in aP2-CCN5 transgenic mice may explain the improvement in insulin production and sensitivity. The increase in GLUT4 protein level would also improve glucose transport and insulin sensitivity. Thus, aP2-CCN5 mice exhibited improved insulin sensitivity and reduced fasting glucose levels, along with an increased GLUT4 levels in fat and muscles. The transgenic mice also had increased levels of metabolically beneficial FAHFAs and PAHSAs, which potentially contributed to the improvements in glucose tolerance and insulin sensitivity.

The findings of systemic CCN5 overexpression were associated with hyperplasia of cardiomyocytes, and white and brown fats [5], which was consistent to viral or transgenic systemic overexpression of CCN5 causing cardiac hyperplasia and increased heart mass [5]. On the other hand, the hyperplasia of cardiomyocytes was unexpected, because a previous cardiomyocyte-specific, local overexpression of CCN5 did not induce either hypertrophy or hyperplasia in the heart [43]. In this new study, hyperplasia of brown adipocytes was verified using bromodeoxyuridine (BrdU) incorporation in vivo and was associated with increased mesenchymal stem cell (MSC) proliferation [5].

Despite having a relatively stable fat mass, aP2-CCN5 mice exhibited smaller s.c. adipocytes and decreased serum leptin levels [5]. The latter is likely a result of the reduced adipocyte size and adiposity. The level of GLUT4 protein, a marker of adipocyte differentiation and a regulator of lipogenesis through carbohydrate-response element-binding protein (ChREBP), was increased in adipose depots along with other markers of adipocyte differentiation (insulin receptor, Bmp4, and other lipogenic genes) [5]. These results do not indicate that TGF-β is involved in adipocyte hyperplasia, as decreased levels of the TGF ligand activin/inhibin A were observed [5]. Instead, the adipocyte hyperplasia was likely induced through the BMP4-regulated canonical WNT pathway [5], further supporting the notion that CCN5 does not inhibit MSC proliferation and differentiation through TGF-β. There was no also difference in Smad2 phosphorylation or activation when CCN5 was overexpressed in mesenchymal stem cell (MSC)-like cells [5].

**In summary**, the adipocyte-specific and systemic overexpression of CCN5 causes an increase in lean body mass, improved insulin sensitivity, hyperplasia of cardiomyocytes, and increased heart mass, but decreased fasting glucose levels. These results are mostly consistent with the opposite phenotypes exhibited in CCN5-knockout mice [4]. Nevertheless, when fed an HFD, both CCN5 knockout and aP2-CCN5 transgenic mice exhibited increases in food intake and in the expression of lipogenic genes (Table 1) [4,5]. Moreover, based on the comparison with the CCN5-knockout model, we expect decreased lipid accumulation in fat tissues, decreased fibrosis in fat and the heart, and/or enhanced cardiac function in aP2-CCN5 mice. Systemic elevation of CCN5 level in circulation may further promote proliferation of other cells, such as pancreatic β-cells, which may also improve metabolic health through additional mechanisms. These important questions remain to be addressed in future studies. Following our demonstration that CCN5 promotes pancreatic β-cell proliferation and survival, the evidence presented here, especially the two genetic manipulations described, has clearly established a crucial role for CCN5 in metabolic regulation. Normal CCN5 expression controls adipocyte proliferation or maturation, adipogenesis, insulin responsiveness, fibrosis, obesity, and heart function.

## Figures and Tables

**Figure 1 ijms-22-13418-f001:**
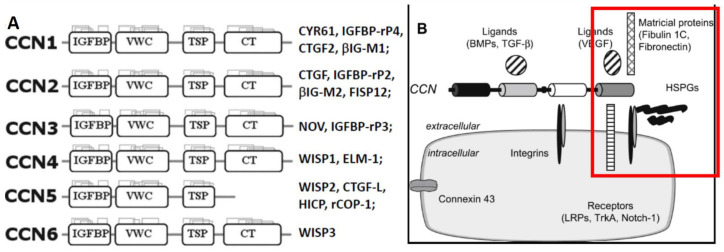
Schematic representation of the CCN/WISP family and partners. (**A**) The domain structure and other names given to different family members. (**B**) CCN proteins are known to interact with a broad range of partners: integrins; ECM components including heparin-sulfated proteoglycans (HSPGs), fibronectin, and fibulin 1C; receptors including low-density lipoprotein receptor-related proteins (LRPs), Notch1, TrkA, and other factors such as bone morphogenic proteins (BMPs), transforming growth factor (TGF)-β, vascular endothelial growth factor (VEGF), S100A4, and connexin 43. The red square indicates what would be missing for the interactions with CCN5 when it lacks the CT domain. Adapted from [7,8,16].

**Figure 2 ijms-22-13418-f002:**
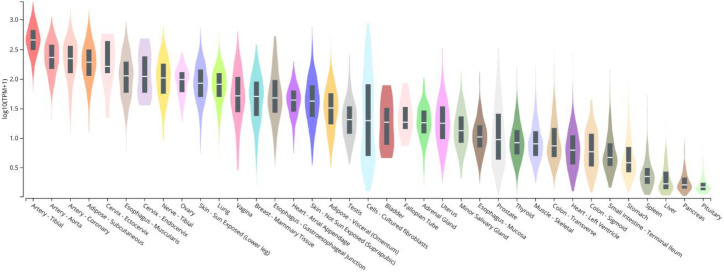
Bulk tissue gene expression for CCN5/WISP2 (ENSG00000064205.10). The Genotype-Tissue Expression (GTEx) project is an ongoing effort to build a comprehensive resource to study tissue-specific gene expression and regulation. Samples were collected from 54 non-diseased tissue sites across nearly 1000 individuals, primarily for molecular assays including WGS, WES, and RNA-Seq. The Y axis represents relative gene expression valued in *log* (TPM) (transcripts per million). Downloaded from: https://www.gtexportal.org/home/, accessed on 9 December 2021).

**Table 1 ijms-22-13418-t001:** Differential phenotype caused by CCN5 gene deficiency vs. adipocyte-specific overexpression, based on two reports and a review [4,5,22].

	CCN5-Knockout	aP2-CCN5
**When fed normal chow diet (NCD or LFD)**		
Body weight	Slight increase	Normal
Adipocyte	Hypertrophy	Normal
Fat mass (obesity)	Increased	Normal
Lean mass		Increased
Insulin sensitivity	Decreased	Increased
Glucose tolerance	Decreased	Increased
Glycemia	Elevated	Normal
Insulinemia	Normal/Elevated	Normal
Water intake	Decreased	
Food intake	Normal	Normal
Energy expenditure		Normal
Cardiomyocyte	Hypertrophy	Hyperplasia
Cardiac function	Systolic + Diastolic deficiency	Not tested
Lipid accumulation	Increased in heart	
Fibrosis	Heart, fat	Not tested
Adipogenic genes	4 Increased (SREBP, C/EBP, PPARγ, aP2)	
Adiponectin level	Decreased	
**When fed high-fat diet (HFD)**		
Body weight	Increased	Slight increase
Adipocyte	Further hypertrophy	sWAT: hyperplasia, hypotrophy
Fat mass (obesity)	Increased	Decreased
Lean mass		Increased
Insulin sensitivity	Decreased	Increased
Glucose tolerance	Decreased/Intolerance	Increased
Glycemia	Higher, Mild diabetes	Decreased
Insulinemia	Elevated/Normal	Decreased
Water intake	Increased	
Food intake	Increased	Increased
Energy expenditure		Increased (per body weight)
Cardiomyocyte	Hypertrophy	Hyperplasia
Cardiac function	Deficient	Not tested
Lipid accumulation	Further increased in heart	
Fibrosis	Heart (more), fat	Not tested
Adipogenic genes	2 Increased (PPARγ, aP2)	13 Increased including SREBP
Adiponectin level	Decreased	Increased

**Table 2 ijms-22-13418-t002:** **Differential roles played by CCN proteins in obesity, fibrosis, and pancreatic islets.** Qualitative roles are summarized based on gain- and loss-of-function experiments; see text in Section 4 for details and original references.

	CCN2	CCN3	CCN4	CCN5
**Adipocyte and adiposity**				
Proliferation		Promote		Promote
Hypertrophy		Increase		Limit
Adipocity		Increase		Decrease
Energy expenditure		Decrease		
**Pancreatic β-cell and insulin secretion**				
β-cell proliferation	Promote, early	Decrease	Promote	Promote
Islet vascularity	Increase			
β-cell maturition	Promote			
Insulin secretion		Impair		
Fibrosis, heart	Pro-	Anti-	Pro-	Anti-

## Data Availability

Not applicable.

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
