# Peer review of "Metabolic Effects of CCN5/WISP2 Gene Deficiency and Transgenic Overexpression in Mice"

_ijms, 2021, doi:10.3390/ijms222413418_

Round 1
Reviewer 1 Report
The review by Tara Alami and Jun-Li Liu, titled “Metabolic effects of CCN5/WISP2 gene deficiency and trans-genic overexpression in mice (a minireview)” explains the potential roles of CCN5/WISP2 gene regulation affects by including multiple aspects (such as obesity, insulin resistance, hyperglycemia, cardiomyopathy, and pancreatic islets) that are somehow related to metabolism in different tissues.
The review is broadly clarifying the importance of the CNN/WISP gene family, which are matricellular proteins and play a pivotal role in different cellular processes including development, proliferation, extracellular matrix remodeling, and also inflammation. Furthermore, also addresses potentially helpful points for future studies.
The role of CCN5 in metabolism and its effects in different tissues, and also, the deficiency (in knock-out mice) and transgenic overexpression of CCN5 through different studying approaches (such as normal chow diet, low- and high-fat diet) is well-documented from physiological- and pathological-views. The comparison of the CNN family such as CNN5, CNN2, CNN3, and CNN4 provides a better understanding of their roles in obesity, fibrosis, and pancreatic islets.
Major comments:
- The abstract is the most important part of the article which lets readers get the gist or essence of the paper. I would like to suggest elaborating the abstract to major aspects of the review. Unfortunately, in the submitted review the abstract is remaining weak, after reading the full paper, it seemed that should be re-considered.
- The knowledgeful information in the texts is very helpful, but the texts-flow should be re-considered to make it more understandable.
- In figure 1, where it represents the CCN/WISP family and partners. In part B of the figure, CCN proteins interact with a broad range of partners could be explained more in detail such as what are these interactions can be addressed under the explanation of the figure.
Author Response
Major comments:
The abstract is the most important part of the article which lets readers get the gist or essence of the paper. I would like to suggest elaborating the abstract to major aspects of the review. Unfortunately, in the submitted review the abstract is remaining weak, after reading the full paper, it seemed that should be re-considered.
Response: We have doubled the length of the Abstract to allow some elaboration on the major findings.
The knowledgeful information in the texts is very helpful, but the texts-flow should be re-considered to make it more understandable.
Response: We have revised (moved around) the section about the heart in Part II and many other parts in order to improve readability.
In figure 1, where it represents the CCN/WISP family and partners. In part B of the figure, CCN proteins interact with a broad range of partners could be explained more in detail such as what are these interactions can be addressed under the explanation of the figure.
Response: We have provided detailed full names for HSPGs, LRPs, BMPs, TGF-β, and VEGF.
Reviewer 2 Report
In this study, Tahar et al reviewed/updated the role of CCN5/WISP2 in metabolic regulation by comparing different experimental findings as a minireview.
I general, I liked the work, but I do have some comments that hopefully will make the work in better shape:
1- The abstract needs to be revised. as it does not sound great!
2- The author is advised to include another table showing the different roles of CCN2, CCN3, and CCN4 in comparison to CCN5.
3- It may be a good idea to include a figure showing the expression profile of CCN5 in different tissues using some public software such as GTX.
Author Response
In this study, Tahar et al reviewed/updated the role of CCN5/WISP2 in metabolic regulation by comparing different experimental findings as a minireview. I general, I liked the work, but I do have some comments that hopefully will make the work in better shape:
1- The abstract needs to be revised. as it does not sound great!
Response: We have doubled the length of the Abstract to allow some elaboration on the major findings.
2- The author is advised to include another table showing the different roles of CCN2, CCN3, and CCN4 in comparison to CCN5.
Response: A new Table 2 has been presented.
3- It may be a good idea to include a figure showing the expression profile of CCN5 in different tissues using some public software such as GTX.
Response: We have retrieved a tissue expression profile for CCN5/WISP2 from GTEx and presented as a new Figure 2.
Round 2
Reviewer 1 Report
I have no further requests
Reviewer 2 Report
I am happy with the revised version.
The author has done all requested comments.